# SIMPLICITY IN COMPLEXITY

**Tingke Shen**[1,*]**, Surabhi S Nath**[1,2,3,*]**, Aenne Brielmann**[2]**, Peter Dayan**[1,2]
[1]Max Planck Institute for Biological Cybernetics, Tübingen, Germany
[2]University of Tübingen, Tübingen, Germany
[3]Max Planck School of Cognition, Leipzig, Germany
{tingke.shen, surabhi.nath, aenne.brielmann, peter.dayan}
@tuebingen.mpg.de

## ABSTRACT

The complexity of visual stimuli plays an important role in many cognitive phenomena, including attention, engagement, memorability, time perception and aesthetic evaluation. Despite its importance, complexity is poorly understood and ironically, previous models of image complexity have been quite *complex*. There have been many attempts to find handcrafted features that explain complexity, but these features are usually dataset specific, and hence fail to generalise. On the other hand, more recent work has employed deep neural networks to predict complexity, but these models remain difficult to interpret, and do not guide a theoretical understanding of the problem. Here we propose to model complexity using segment-based representations of images. We use state-of-the-art segmentation models, SAM and FC-CLIP, to quantify the number of segments at multiple granularities, and the number of classes in an image respectively. We find that complexity is well-explained by a simple linear model with these two features across six diverse image-sets of naturalistic scene and art images. This suggests that the complexity of images can be surprisingly simple.

## 1 INTRODUCTION

The subjective complexity of sensory stimuli plays an important role in many cognitive phenomena, including attention, engagement, memorability, time perception or aesthetic evaluation Kyle-Davidson & Evans (2023); Sun & Firestone (2021); Van Geert & Wagemans (2020); Palumbo et al. (2014), and is relevant to a wide range of real-world applications such as advertising, web design, and computer graphics Pieters et al. (2010); Wu et al. (2016); Reinecke et al. (2013); King et al. (2020); Ramanarayanan et al. (2008). It is therefore important to understand the factors and mechanisms underlying the perception of complexity. Most empirical and theoretical work concerns artificial or naturalistic images Chikhman et al. (2012); Gartus & Leder (2017); Nath et al. (2023); Machado et al. (2015); Nagle & Lavie (2020); Guo et al. (2023); the latter are the focus of our work.

There is by now a range of datasets containing human ratings of the complexity of various subcategories of naturalistic images—we consider *RSIVL* (*RSIVL-RS1*) Corchs et al. (2016), *VISC* (*VISC-C*) Kyle-Davidson et al. (2023), *Savoias* Saraee et al. (2020) and *IC9600* Feng et al. (2022). Duly, there has then been a number of attempts to predict these ratings, and thereby understand the computations concerned. Note, though, that these methods have hitherto largely been applied on their own, separate, datasets, rather than being directly compared. The methods fall into two broad categories: using either simple (often linear) combinations of handcrafted image features, or modern convolutional neural networks (CNNs) as predictors or feature extractors. We advocate a middle ground, revealing an unexpected degree of simplicity in modelling complexity.

For the first category of methods, several qualitative and quantitative image features have been proposed and shown to predict complexity. These include the number and variety of elements, colour, edge density, file size, Fourier slope, HOG and information-theoretic measures such as entropy and information gain Van Geert & Wagemans (2020). Corchs *et al.* compiled 11 measures based on spatial, frequency and color properties which were combined linearly to fit perceived complexity ratings on the *RSIVL* dataset. They found the number of regions, frequency factor and number of colours received the largest weights Corchs et al. (2016).

Equally, Kyle-Davidson *et al.* proposed measures of clutter (see also Olivia et al. (2004); Rosenholtz et al. (2007); Fan et al. (2017)), entropy and patch-wise symmetry as determinants of complexity, showing good performance on the *VISC* dataset.

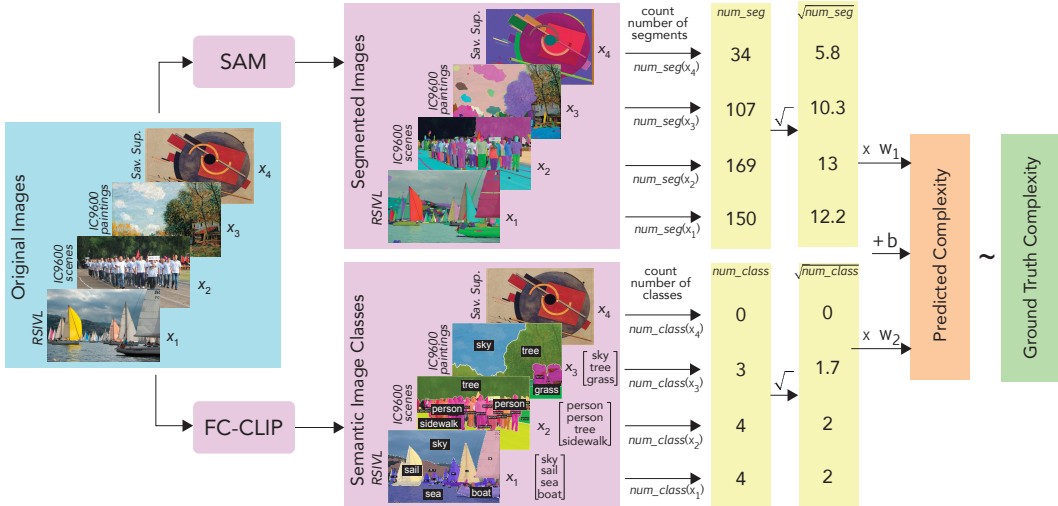

Figure 1: Overview of methods. Our complexity model is shown. Images from across 8 different scenes and art image-sets are passed through 2 segmentation models–SAM, for segmentation, and FC-CLIP for semantic segmentation. Example images are shown for 4 image-sets, namely *RSIVL*, *IC9600 scenes*, *IC9600 paintings* and *Savoias Supremantism* (*Sav. Sup.*). The outputs of SAM are shown as Segmented Images, where the detected segments are highlighted, and the outputs of FC-CLIP are shown as Semantic Image Classes where the image with detected classes and a list of classes obtained are shown. For clarity, only a subset of classes detected by FC-CLIP are shown in each image. The predicted segments and class-instances from SAM and FC-CLIP are counted and the counts are deemed *num_seg* and *num_class*. These two features are then transformed using square root function. The resulting $\sqrt{num\_seg}$ and $\sqrt{num\_class}$ features are linearly combined to estimate complexity.

The advantage of hand-crafted features is that they are largely interpretable. However, they are often dataset-specific, possibly due to the difficulty of evaluating such rather subjectively-defined measures in general. Perhaps as a result, a large number of these potentially noisy features seem to be required to predict subjective complexity well.

More recently, it has become popular to exploit the computational capabilities of deep neural networks to extract relevant image features. Analysis on *Savoias* dataset, comprising of 1400 images across 7 categories showed that activations from intermediate layers of a CNN pretrained on object or scene recognition correlated best with human complexity ratings Saraee et al. (2020). These authors also compared unsupervised and supervised methods, suggesting that supervision can improve prediction.

Feng and colleagues built further on this work, first by introducing a large-scale visual complexity dataset comprising on 9600 images across 8 semantic categories, and then providing a CNN-based method predicting scores and activation maps Feng et al. (2022). This model achieved high test performance, outperforming previous methods.

However, although such CNN-based models perform well, and can even generalise competently to unseen images, they are hard to interpret (as activation maps do not convey much information, and can also be unreliable Bilodeau et al. (2024)) and do not guide a theoretical understanding of the problem.

Here, we benefit from both categories of methods. We use modern foundation models Bommasani et al. (2021) to evaluate particular hand-crafted features in a way that generalizes across many classes of images. We then combine these features linearly to predict complexity.

To choose hand-crafted features, we start from the observation that features that fragment images in meaningful ways tend to estimate complexity relatively well (for example, clutter in Kyle-Davidson et al. (2023) or the number of regions in Corchs et al. (2016)). We therefore leverage the capabilities of state of the art (SOTA) image segmentation models to extract relevant segments from the image at multiple spatial granularities. Such models are the closest existing approximations to how humans represent scenes for two main reasons: first, the models are trained using a vast amount of annotations from several humans and hence reflect relevant inductive biases, and second, the architecture of CNNs and transformers are inspired by the human visual processing systems and generalize surprisingly well to unseen images. With the help of such models, we obtain semantically consistent segments at different spatial granularities relevant to perceptual image processing Epstein & Baker (2019). We then derive from them the core components of perceived complexity.

With improved quality of feature extraction and evaluation, we make the central observation that only few features are necessary to predict complexity well, justifying the claim that complexity can be surprisingly simple.

## 2 METHOD

We develop a parsimonious model of the perceived complexity of naturalistic images using two types of segmented features, namely the number of segments, and the number of named classes, extracted from SOTA segmentation models. Our method is described in Figure 1. We also use an additional measure called patch-symmetry (borrowed from Kyle-Davidson et al. (2023)) to address a main failure mode of our model.

### 2.1 DATASETS

We use 4 freely available naturalistic image datasets with corresponding subjective complexity ratings, namely *RSIVL* Corchs et al. (2016), containing 49 scene images; *VISC* Kyle-Davidson et al. (2023), containing 800 scene images across 12 sub-categories, *Savoias* Saraee et al. (2020), containing 1400 images across 7 categories, and *IC9600*, containing 9600 images across 8 categories Feng et al. (2022). We use the mean subjective complexity per image across raters (there were between 10 to 26 raters per image across datasets) as ground truth. We restrict to scenes and art image categories and omit advertisement (*Savoias* and *IC9600*) and visualisation (*Savoias*) categories since they contain substantial amounts of text. We combine similar image categories within a dataset to generate 8 image-sets for analysis: (1) *RSIVL*, containing all *RSIVL-RS1* images; (2) *Savoias Scenes* (*Sav. Scenes*), comprising of *Savoias* scene and object categories; (3) *IC9600 Scenes* (*IC9. Scenes*), comprising of *IC9600* scene, object, person, transportation and architecture categories; (4) *Savoias Art* (*Sav. Art*); (5) *Savoias Suprematism* (*Sav. Suprematism*); (6) *IC9600 Paintings* (*IC9. Paintings*); (7) *VISC*, containing all *VISC-C* images, and lastly (8) *Savoias Interior Design* (*Sav. Int*), which is considered separately as it contains software-generated 3D-rendered images.

### 2.2 FINDING SEGMENTS USING A FOUNDATIONAL SEGMENTATION MODEL

We extracted segments in images using the SOTA Segment Anything Model (SAM) Kirillov et al. (2023). SAM detects blobs of segments in an image at different scales. SAM was trained on the largest public segmentation dataset to date, is capable of zero-shot generalization, and achieves SOTA performance. Based on pilot studies, we set the spatial granularity parameter *points-per-side* to $64$. This allowed the network to find finer segments, and correlated well with ground truth complexity. We set all other parameters of SAM to their default values and evaluated the total number of detected segments per image (*num_seg*).

### 2.3 FINDING CLASSES USING OPEN-VOCABULARY SEMANTIC SEGMENTATION

We found the nameable class instances in an image using FC-CLIP Yu et al. (2023). FC-CLIP is an open-vocabulary panoptic segmentation algorithm that can find multiple instances of each class, and achieves SOTA performance Yu et al. (2023). We use panoptic semantic segmentation to predict classes because multi-scale methods like Semantic SAM Chen et al. (2023) produced many false

positives. We set all parameters of FC-CLIP to default and evaluated the number of detected classes (including repeated classes) per image (*num_class*).

Table 1: Model performance on 6 image-sets and comparison with previous models. The models from previous work are classified as being based on either handcrafted features, or Convolutional Neural Networks (CNNs). * for supervised methods indicate their own *test* set. Bold indicates the best model.

| Model | RSIVL | Sav. Scenes | IC9. Scenes | Sav. Art | Sav. Suprematism | IC9 Paintings |
|---|---|---|---|---|---|---|
| **Handcrafted features** | | | | | | |
| Corchs 1 (10 features) | 0.66 | 0.62 | 0.70 | 0.68 | 0.80 | 0.53 |
| Corchs 2 (3 features) | 0.77 | - | - | - | - | - |
| Kyle-Davidson 1 (2 features) | 0.68 | 0.54 | 0.54 | 0.55 | 0.79 | 0.49 |
| **CNNs** | | | | | | |
| Saraee (transfer) | 0.72 | 0.67 | 0.59 | 0.55 | 0.72 | 0.58 |
| Kyle-Davidson 2 (supervised) | 0.50 | 0.36 | 0.41 | 0.30 | 0.15 | 0.33 |
| Feng (supervised) | 0.83 | **0.79** | **0.94*** | **0.81** | 0.84 | **0.93*** |
| **Our method** | | | | | | |
| $\sqrt{num\_seg}$ | 0.78 | 0.65 | 0.81 | 0.67 | 0.89 | 0.82 |
| $\sqrt{num\_class}$ | 0.70 | 0.75 | 0.73 | 0.56 | 0.27 | 0.67 |
| $\sqrt{num\_seg} + \sqrt{num\_class}$ | **0.83** | 0.78 | 0.84 | 0.73 | **0.89** | 0.83 |

Intuitively, FC-CLIP finds the most salient, lower granularity semantic classes in the image while SAM finds sub-components of these classes at higher granularities. As a result, *num_seg* is larger than *num_class* for all images.

## 2.4 Linear Regression Model

We estimate subjective complexity using multiple linear regression. A preliminary examination showed that subjective complexity scales roughly linearly with $\sqrt{num\_seg}$ and $\sqrt{num\_class}$, hence, we apply a square-root transformation to our features. We used the `statsmodels` OLS function in Python to fit multiple linear regression on each image-set. We perform 3-fold cross-validation $M$ times, where $M$ is larger for smaller image-sets, and report the average Spearman correlation over all *test* sets. We compare our models to six baselines from previous work. These baselines include three handcrafted feature-based baselines—two from Corchs et al. (2016)–Corchs 1, comprising of their 3 best features M8, M5 and M10 (only tested on RSIVL since we were unable to implement M8 (*number of regions*) to apply it for other datasets), Corchs 2 comprising of 10 features M1 to M11 (excluding M8) and one baseline from Kyle-Davidson et al. (2023) comprising of their clutter and patch-wise symmetry measures. The other three are CNN baselines, namely the supervised method from Kyle-Davidson et al. (2023), the transfer-learning method from Saraee et al. (2020) and the supervised method from Feng et al. (2022).

## 3 Results

### 3.1 Excellent performance on natural scenes and art

Table 1 shows the performance of our models and baselines for 6 image-sets. We see that our linear model with $\sqrt{num\_seg}$ and $\sqrt{num\_class}$ attains a Spearman correlation between 0.73 to 0.89 with human complexity judgments across natural scenes and art image-sets. Notably, our model performs better than all handcrafted feature baselines, the transfer-learning neural network method from Saraee et al. (2020) and the supervised neural network from Kyle-Davidson et al. (2023).

It performs similarly to the supervised neural network from Feng et al. (2022). The exceptions are the test datasets from the same paper and *Savoias Art*. This neural network *directly* learns a high-dimensional mapping from image to complexity, thereby discovering features that best predict complexity in a supervised way. We show that in many cases, this high-dimensional relationship can

be distilled down to simply the number of segments and named instances in the image. Hence, we provide evidence that complexity is computable from segmentation features, rather than requiring features that are explicitly optimized for complexity.

We also compared the full model with versions restricted to just one of the $\sqrt{num\_seg}$ or $\sqrt{num\_class}$ terms. We see that both terms contribute to the variance explained for all datasets except *Savoias Suprematism*, where $\sqrt{num\_class}$ fails to explain additional variance on top of $\sqrt{num\_seg}$. This is because *Savoias Suprematism* contains abstract art images with geometric shapes, and FC-CLIP fails to find appropriate nameable classes as in its training set. However, for *Savoias Suprematism*, the model with only $\sqrt{num\_seg}$ already explains high variance and achieves performance superior to all other models, suggesting that the number of segments at multiple granularities drives perceived complexity in images composed of geometrical shapes that lack overt semantics (at least for art-novice raters).

Figure 2 shows the images with the highest and lowest predicted complexity from each of the 6 image-sets in Table 1. The highest predicted images are those with many entities and hence high $\sqrt{num\_seg}$ and $\sqrt{num\_class}$. The lowest predicted images have only a few entities and hence low $\sqrt{num\_seg}$ and sometimes zero $\sqrt{num\_class}$.

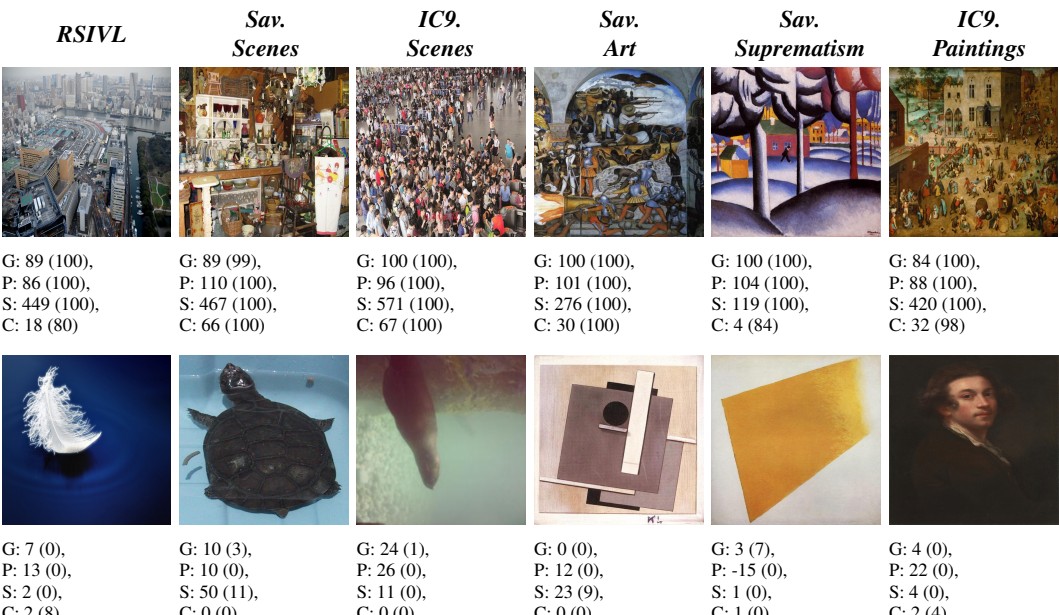

Figure 2: Images with the highest (top row) and lowest (bottom row) complexity predictions for the 6 image-sets in Table 1. G = ground truth complexity from 0 to 100, P = predicted complexity, S = *num_seg*, C = *num_class*. Percentiles of the corresponding values are shown in brackets. The highest predicted images have many entities and hence high *num_seg* and *num_class*. The lowest predicted images have only a few entities and sometimes zero *num_class*.

Figure 3 shows the mean and standard deviation of the ground truth subjective complexity for images for each bin of $\sqrt{num\_seg}$ and $\sqrt{num\_class}$ on an example image-set, *IC9600 Scenes*. In general, and as expected, mean ground truth complexity increases with increasing $\sqrt{num\_class}$ and increasing $\sqrt{num\_seg}$ (also well-matched to predictions) showing that a complex image is one with both a large number of segments and classes. The standard deviation of the subjective complexity, which contributes markedly to the prediction error, is particularly high in the bins with the greatest and least $\sqrt{num\_class}$. This suggests that FC-CLIP might over- or under-predict classes. The largest discrepancies lie in the bin with the highest $\sqrt{num\_seg}$ and the lowest $\sqrt{num\_class}$. Here, FC-CLIP often fails to find any nameable segments at all.

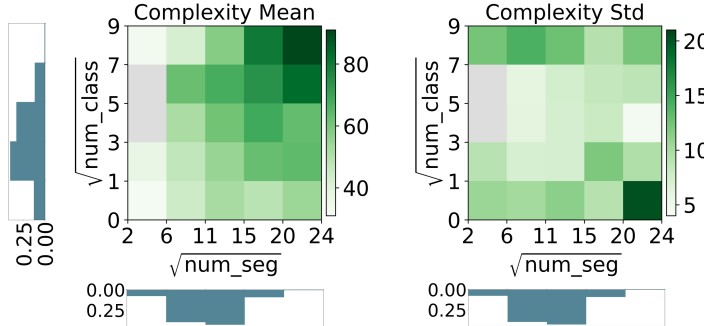

Figure 3: Mean and standard deviation of the ground truth subjective complexity in different bins of $\sqrt{num\_seg}$ and $\sqrt{num\_class}$ for *IC9600 Scenes*. Means show that complexity increases with $\sqrt{num\_seg}$ and $\sqrt{num\_class}$. Standard deviations reveal that FC-CLIP might over- or under-predict classes.

## 3.2 FAILURE MODE: SYMMETRY AND STRUCTURE

The statistics of segments and classes at multiple granularities explain most of the variance in the datasets we tested. However, the structure in the image, *i.e.*, the spatial and functional relationships between elements is also known to be an important contributor to complexity (Chipman (1977); Ichikawa (1985), Gestalt theory of perception). Indeed, we find that segment statistics alone are not enough to adequately explain complexity judgments in two other image-sets: *VISC* and *Savoias Interior Design*. Figure 4 shows an example from each dataset with the highest prediction errors. In each case, our model over-predicts complexity because SAM finds too many segments without accounting for the fact that many segments are arranged in a spatial pattern (books in the top row and windows in the bottom row). Further, *num_class* does not contribute to reducing complexity in such cases, either because it is also high (since the uniqueness of classes is not accounted for), or because the weight of the *num_class* term is learned to be low (for example in *VISC*) We see that both of these images have high patch-symmetry (defined in Kyle-Davidson et al. (2023)). Figure 5 illustrates a significant, positive correlation between patch-symmetry and model error (prediction minus ground truth), showing that our model tends to over-predict when the image is more spatially symmetric, *i.e.* has more spatial structure. Table 2 shows that when *patch_symmetry* is added as a feature to the regression, our model improves in Spearman correlation by atleast 0.12, and becomes competitive with most baselines.

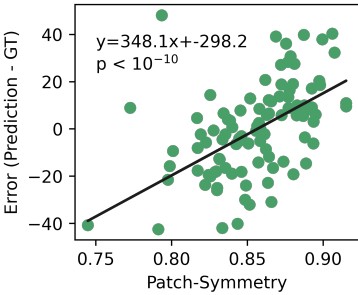

Figure 5: The relationship between patch-symmetry and prediction error for *Sav. Int.* Model overpredicts when patch-symmetry is high. Linear regression reveals a significant Pearson correlation of 0.51.

| Model | VISC | Sav. Int. |
|---|---|---|
| **Handcrafted features** | | |
| Corchs 1 (10 features) | 0.62 | 0.85 |
| Kyle-Davidson 1 (2 features) | 0.60 | 0.74 |
| **Neural network** | | |
| Saraee (transfer) | 0.58 | 0.75 |
| Kyle-Davidson 2 (supervised) | - | 0.56 |
| Feng (supervised) | **0.72** | **0.89** |
| **Our method** | | |
| $\sqrt{num\_seg} + \sqrt{num\_class}$ | 0.56 | 0.61 |
| $\sqrt{num\_seg} + \sqrt{num\_class} + patch\_symm$ | 0.68 | 0.80 |

Table 2: Model performance on *VISC* and *Savoias Interior Design* (*Sav. Int*) datasets with and without the *patch_symmetry* feature, and comparison with previous models. Bold indicates the best model. When *patch_symmetry* is added to the regression, our model improves in performance.

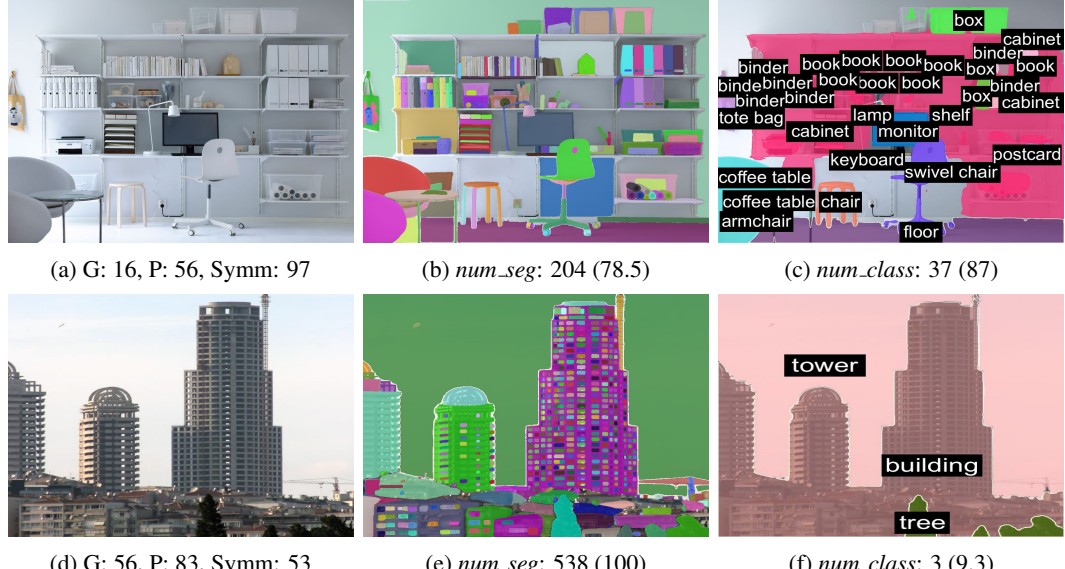

(a) G: 16, P: 56, Symm: 97      (b) *num_seg*: 204 (78.5)      (c) *num_class*: 37 (87)

(d) G: 56, P: 83, Symm: 53      (e) *num_seg*: 538 (100)      (f) *num_class*: 3 (9.3)

Figure 4: Example image with one of the highest prediction errors from the *VISC* (top row) and *Sav. Int* (bottom row) datasets. From left to right: original image, SAM output, FC-CLIP output. G = ground truth complexity from 0 to 100. P = predicted complexity. Symm = patch-symmetry percentile. Percentiles of *num_seg* and *num_class* are also shown in brackets. SAM finds too many segments and without accounting for structure this leads to overprediction. In the top image, FC-CLIP also finds high *num_class*. In the bottom image *num_class* is low but *num_class* does not contribute significantly to the regression for *VISC*. However, both images have high patch-symmetry.

## 4   DISCUSSION

We presented a linear model of complexity using two features extracted using SOTA segmentation neural networks: *num_seg* and *num_class*. Our model outperforms most baselines achieving a Spearman correlation between 0.73 to 0.89 with subjective complexity ratings across six tested image-sets of naturalistic scenes and art. As a result, our model provides a simple explanation of perceived complexity that generalizes across multiple domains and image types. Our results suggest that segment-based representations are good proxies for the cognitive processes underlying human judgments of complexity, a result that could be extended to attention, memorability, aesthetic evaluation, *etc*.

Our model performs better than all handcrafted feature-based baselines. A possible reason for this is that *num_seg* is a significant improvement over previously suggested features for image fragmentation (such as "number of regions" from Corchs et al. (2016)) that approximate the segments in an image. Further, to our best knowledge, we are the first to exploit named-segments corresponding to semantic classes to predict complexity, whose count provides an estimate of the number of lower granularity segments in an image.

Importantly, the segments and classes are both computed by neural networks trained on large datasets of human annotations. Therefore, the predictions are likely to be semantically meaningful, reflecting not only pixel information but also the annotator's prior experiences with the contents of the images in the training set. The annotations and hence segments also encompass multiple levels of spatial and semantic granularity, capturing contributions to complexity across scales. This is in contrast to past works that have tried to approximate spatial granularity and semantic variety using only sliding windows or pyramid scaling of filters Corchs et al. (2016); Kyle-Davidson et al. (2023; 2022); Guo et al. (2023).

Our model performance was generally comparable to the supervised neural network of Feng et al. (2022), which was trained on a large dataset to directly predict complexity. The difference in performance can be explained by the fact that the neural network potentially utilizes many more than two

features and conditionally chooses those features based on the context and distribution of images. However, we have shown that only two features can explain complexity equally well on multiple datasets and domains, elucidating a simpler view of complexity.

In addition, unlike the CNN models, our model is highly interpretable. As we demonstrate in Figure 4, we can attribute predictions or diagnose failure cases by visually inspecting the outputs of SAM and FC-CLIP. Also, the contributions of the segments and classes to the complexity score can be clearly elucidated (as the square root of their counts).

However, our model has limitations. The accuracy of our model depends on the accuracy of the segments and classes predicted by SAM and FC-CLIP. Currently, SAM is incapable of detecting thin, "one-dimensional" patterns. FC-CLIP sometimes misses salient classes or repeated classes (failing to predict *any* classes for some images outside its training distribution, *e.g.* images in *Sav. Suprematism*) and doesn't predict nested classes at multiple granularities (*e.g.* both the "house" and its "window"). As the SOTA segmentation models improve, we expect the performance and interpretability of our model to also increase further.

We also addressed the inability of *num_seg* and *num_class* to account for structure in an image which reduces perceived complexity. We saw that adding *patch-symmetry* to the regression led to competitive performance on *VISC* and *Sav. Int* image-sets. However, as part of future works, we aim to build a more parsimonious model using a segment-based feature of structure. For example, scene-graphs Chang et al. (2021) or generative programs Sablé-Meyer et al. (2022) can be used to organize the named entities detected by FC-CLIP by their spatial and semantic relationships, and image complexity can be derived from the complexity of these representations, for example as their compressibility Mahon & Lukasiewicz (2023); Dehaene et al. (2022); Karjus et al. (2023).

We modeled subjective complexity ratings which was the mean rating across multiple raters. However, complexity judgments are known to vary across both individuals or groups (age, cultures, etc.) Gartus & Leder (2017). For the art datasets, the complexity ratings were given by art-novices and would likely differ significantly from ratings of art-experts Pihko et al. (2011); Bimler et al. (2019). These individual differences could be caused by differences in the segments people perceive (the regions of an image they consider to be part of the same segment). For example, different individuals may segment at different granularities. Individual differences can also be caused by the mapping from the perceived segments to complexity. Explicitly accounting for individual variability using subject-specific data (for example by fine-tuning the regression or segmentation models) will be an important part of future work.

In conclusion, we develop a parsimonious and interpretable account of human perceived complexity in naturalistic images using segmentation-based methods, showing that complexity can be surprisingly simple given the right image representations.

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
