# OpenReview forum: "Simplicity in Complexity"
_ICLR.cc/2024/Workshop/Re-Align — ICLR 2024 Workshop Re-Align Poster_

### Official Review · Reviewer_TK1E · 2024-02-23
**Interesting finding suggests simple explanation for visual complexity, more connection to the workshop theme might be needed**

**Rating:** 1
**Fit:** 2
**Confidence:** 2

**Workshop Review:**

Summary:
This paper investigates an important question, namely the factors and mechanisms underlying subjective complexity of sensory stimuli. The paper offers a very simple explanation to model visual complexity, namely segment-based representations of images. They show complexity of visual stimuli (human ratings) can be well predicted by a simple linear model using information such as number of classes and number of segments extracted from segment-based representations using segmentation models such as SAM and FC-CLIP. A method falls in between using combinations of handcrafted image features and using uninterpretable feature maps out of convnets.

I thought the finding is novel in that it suggests segment-based representations might be good proxies for the cognitive processes underlying human judgments of complexity. However, what prevents me recommending the paper in its current form from acceptance at ReAlign is its lack of clear engagement to the central theme of the workshop, namely the alignment between artificial and biological systems. I believe the finding itself is of relevance as it helps to understand what sort of representations could be at play under perception of complexity in the human brain, I believe more discussion is needed as to what implications are there in terms of understanding alignment.
I find some of the motivations and justifications for the choice of the SOTA segmentation models to understand biological representations a little confusing and probably need more clarification. For instance, I am not sure if SOTA segmentation models like SAM can be categorized as “the closest existing approximations to humans represent scenes” for using CNNs/Transformers underneath. Yes CNNs are shown to correspond to human visual systems but transformers are not and certainly not inspired by visual processing. More importantly, SAM underneath is a mixture of ViT, CLIP and transformer decoders which is a beast at creating good image segments, but it feels a bit of a stretch to me that it is justified to say SAM aligns with how humans perceive visual scenes and create segments. In fact, doesn’t it suggest the opposite that so long as you have a powerful segmentation model to produce good segments, they can be linearly combined to predict complexity ratings? We probably need evidence (not suggesting it should be done in the current work!) supporting that segments like these are indeed present in biological representations in the brain. But I don’t want to take the credit away from this work. It has certainly offered an interesting start in this direction.

I find the writing of the paper could be improved especially for readers who aren’t neck deep into the exact tasks, models and datasets presented in this work. High-level overviews of the baseline models for instance could be expanded a little more to give readers more intuitions as to what the authors are comparing against. At times I find the results are not clearly connected to figures. For instance, “We show that in many cases, this high-dimensional relationship can be distilled down …” -- where is the figure supporting this? Though the result as stated itself is interesting.

**Reason For Not Giving Higher Score:**

I could not give a higher score due to the paper as is is lacking clear engagement to the central theme of this workshop. Some discussions on the Indications of the results to aligning bio-aritificial systems could be added.

**Reason For Not Giving Lower Score:**

N/A

**Reviewer Domain:**

cognitive science

---

### Official Review · Reviewer_o896 · 2024-02-23
**Review: Simplicity in Complexity**

**Rating:** 2
**Fit:** 2
**Confidence:** 2

**Workshop Review:**

The authors of this paper introduce a novel method for assessing human-perceived complexity in naturalistic images. This approach employs an image segmentation model, specifically SAM, and an open-vocabulary object detection model, FC-CLIP. The authors demonstrate that a linear combination of the square root of detected segments and the square root of detected classes can serve as a measure for image complexity. The authors show that this approach performs exceptionally well, even achieving top performance on several datasets. A key feature of the proposed approach is its interpretability, as the estimated score can always be traced back to the number of found segments or number of classes.

 - *Clarity*: The paper is well-structured and easy to comprehend. The authors provide a comprehensive overview of related works, and the description of the methods is lucid.
 - *Correctness*: The paper’s claims are substantiated by evidence from several experiments and comparisons.
 - *Novelty*: The proposed method exhibits novelty.
 - *Interest to the Community*: The proposed approach demonstrates that human-perceived complexity can be estimated using image segmentation and object detection models. The authors also address the role of symmetry and structure in human complexity perception. Although this work is somewhat outside the general scope of representation alignment, I believe it will be of interest to the community.

A few comments:
1. It would be beneficial to report the impact of the points-per-side parameter in the SAM model, as it limits the number of potential segments. Given that this might be crucial for image complexity estimation, it would be interesting to observe the changes in the performance of the proposed approach when varying this parameter.
2. For the estimation of the number of classes, are only unique classes counted, or are classes for all detected objects included?
3. From the writing, it remains somewhat unclear whether the linear regression model is retrained on each dataset or is general across all datasets. Additionally, I believe it is important to state the linear coefficients to illustrate which term is more significant to the complexity estimation.

**Reason For Not Giving Higher Score:**

While the work presented in the paper is indeed intriguing and introduces novel concepts, it appears to deviate slightly from the central theme of analyzing the alignment between human and artificial representations.

**Reason For Not Giving Lower Score:**

Discussed in review.

**Reviewer Domain:**

machine learning

---

### Official Review · Reviewer_wpQj · 2024-02-25
**Ignored complexity of model, simplicity in insight**

**Rating:** 2
**Fit:** 3
**Confidence:** 3

**Workshop Review:**

The authors critique the inadequacy and dataset-specific nature of handcrafted features, as well as the opacity of deep neural network models that have been previously used to gauge complexity in pursuit of developing a simpler model. They propose, what they describe as a a simpler approach, using segment-based representations through advanced segmentation models—SAM and FC-CLIP—to evaluate image complexity. By quantifying the number of image segments and classes, they demonstrate that a simple linear model can effectively capture complexity across varied sets of natural and artistic images, challenging the notion that understanding image complexity requires complex models.

This is an interesting attempt to operationalize "visual complexity". The author's approach is fairly simple, but uses very complex segmentation algorithms. Saying that the new model is simple (given they are derived from SOA SAM and FC-CLIP, which are highly complex) is rather misleading. I am guessing that simpler segmentation models do not do as well? This would be a direction to explore further.

The central insight that I drew from this work is that whatever human participants mean by complexity can be approximated by the ~number of objects (and segments) present in an image.

Another interesting direction could be the development of better evaluation benchmarks of visual complexity (decorrelated from number of objects) leveraging the failure models of the current models.

I do appreciate the authors delving deeper into their method and identifying failure models (Figure 4).

**Reason For Not Giving Higher Score:**

Rating wise, I think this a good paper for the meeting. But it does not present a particularly novel approach to warrant a talk? I did enjoy reading the study.

**Reason For Not Giving Lower Score:**

Rating wise, I think this a good paper for the meeting. But it does not present a particularly novel approach to warrant a talk? I did enjoy reading the study.

**Reviewer Domain:**

neuroscience

---

### Decision · Program_Chairs · 2024-03-02

Accept (Poster)